# Shortened PGLYRP1 Peptides Regulate Antitumor Activity of Cytotoxic Lymphocytes via TREM-1 Receptor: From Biology to Bioinformatics

**DOI:** 10.3390/ijms26094069

**Published:** 2025-04-25

**Authors:** Daria M. Yurkina, Kirill A. Shcherbakov, Elena A. Romanova, Anna V. Tvorogova, Alexey M. Feoktistov, Georgii P. Georgiev, Denis V. Yashin, Lidia P. Sashchenko

**Affiliations:** 1Institute of Gene Biology (RAS), 119334 Moscow, Russia; yrkina121@gmail.com (D.M.Y.); kirill.soff@gmail.com (K.A.S.); elrom4@rambler.ru (E.A.R.); annatvor@mail.ru (A.V.T.); a.feo95@mail.ru (A.M.F.); georgiev@genebiology.ru (G.P.G.); sashchenko@genebiology.ru (L.P.S.); 2Institute of Molecular Biology (RAS), 119334 Moscow, Russia

**Keywords:** PGLYRP1, TREM-1, cytotoxicity, peptides, molecular docking, antitumor activity, receptor activation

## Abstract

The pro-inflammatory immune response plays an important role in protecting the body from pathogens and tumors. In this study, we were able to identify three peptides of the innate immunity protein PGLYRP1 (Tag7) that could regulate the activity of the TREM-1 receptor. TREM-1 receptor activation on monocytes triggers the appearance of antitumor lymphocytes. All three peptides studied (17.0, N9, and N15) bind with the TREM-1 receptor with the Kds 1.32 ± 0.2 nM, 9.66 ± 0.5 nM, and 7.43 ± 0.4 nM, respectively. An N9 peptide inhibiting the activity of the receptor was identified in addition to two peptides (N9 and N15) that jointly trigger the activation of the receptor. The conducted molecular docking study revealed amino acid residues (Ile57, Ile58, Glu106, Ser108, Leu110, Tyr116, Pro118, Pro119, Arg130, and Val 132), necessary for various functions of peptides, providing important knowledge for understanding the mechanism of activation of this receptor that can also serve as a basis for the development of therapeutic drugs to regulate its activity in the treatment of autoimmune diseases and tumors.

## 1. Introduction

The study of the interactions between ligands and pro-inflammatory receptors is the subject of extensive research in modern immunology. A wide range of pro-inflammatory receptors have been described in the literature, including TLRs, RAGE, TNFR1, and TREM-1 [1,2,3,4,5,6]. In 2000, a new, crucial immunomodulator for the development of inflammatory processes, TREM-1 (Triggering receptor expressed on myeloid cells 1) innate immunity receptors, was described in the literature [6]. This receptor is present on the surface of immune cells, including monocytes, neutrophils, and dendritic cells, and, to a lesser extent, NK cells and some subpopulations of T and B lymphocytes [7,8]. TREM-1 is involved in the induction of pro-inflammatory cytokine gene expression and the regulation of T cell proliferation, activation of antigen-presenting cells, and antivirus protection [9,10]. In synergy with TLR4, it is responsible for the overexpression of inflammatory cytokines (“cytokine storm”), leading to increased development of inflammation and irreversible consequences for the body [11]. Overactivation of TREM-1 is associated with sepsis, arthritis, colitis, and viral diseases, including COVID-19 and Ebola [12,13,14,15,16,17,18,19]. A wide range of TREM-1 ligands has been described. It has been shown that the stimulation of inflammatory processes in mice depends on the interaction of the TREM-1 receptor with actin and RNA-binding protein [20,21,22]. Viral proteins can also interact with TREM-1. It has been shown that the surface glycoprotein (SR) of Ebola virus (EBV) and Marburg virus (MARV) can act as TREM1 ligands [18]. The ligand of TREM-1 also includes multifunctional proteins, HMGB1 [23,24,25], the main heat shock protein Hsp70 [26,27,28], and the innate immunity protein PGLYRP1 [29,30]. PGLYRP1 has been identified as an innate immunity protein that recognizes the peptidoglycan of the bacterial wall and participates in defense against microorganisms in insects and mammals [31,32,33]. The gene in mice was first discovered in the IBG RAS [34,35], with the protein expressed being designated Tag7. The antitumor activity of this protein has been studied [36].

The interaction of ligands with TREM-1 on the cell membrane leads to dimerization of the receptor, followed by activation of the metalloprotease MMP-9 [37], cleavage, and dissociation of the exodomain from the cell surface. sTREM-1 is considered a biomarker of infection and viral diseases [38,39]. Further activation of the receptor develops in two ways. Triggering cascades of tyrosine kinases leads to activation of the NFkß factor, expression of pro-inflammatory cytokines, and development of inflammatory processes [40,41]. In turn, the release of cytokines leads to the appearance of cytotoxic lymphocytes. We have shown that three TREM-1 ligands, Tag7, Hsp70, and HMGB1 proteins, not only stimulate the production of pro-inflammatory cytokines but also induce the activation of cytotoxic lymphocytes against MHC-negative tumor cells [24,27,42].

The identification of biologically active epitopes of regulatory proteins represents a leading trend in modern immunology. Shortened peptides that interact with receptors contribute to understanding the interaction between receptors and ligands. Inhibitory peptides can be used in immunotherapy. In their study, G. Multhoff isolated a fragment of the Hsp70 protein, the 14-membered TKD peptide, which is responsible for various functions of this protein [43]. TNF inhibitory and activating peptides have also been described in the literature [44,45]. The seven-membered peptide eCIRP (M3) significantly reduced TREM-1-dependent inflammation [46]. We have identified Tag7 peptides that inhibit or activate TNFR-1-dependent cytotoxic signal [47]. We also identified 10-membered peptide 17.0 and 24-membered peptide N3 located in the N-terminal region of the Tag7 polypeptide chain. Peptide 17.0 inhibited the activation of TREM-1 under the action of Tag7 [48]. The N3 peptide fully reproduces the function of full-size Tag7 and activates the TREM-1 receptor [49].

In this work, we focused on the identification of shortened fragments of the N3 peptide and the study of their effect on the cytotoxic activity of lymphocytes. The primary aims of our study were as follows: (1) to identify and synthesize shortened fragments of the N3 peptide; (2) to investigate the inhibitory activity of these peptides; (3) to investigate the ability of these peptides to stimulate the cytotoxic effect of lymphocytes; and (4) to study the interaction between these peptides and TREM-1 using a mathematical model.

## 2. Results

### 2.1. Peptides N9 and N15 Do Not Induce Cytotoxicity of Lymphocytes

First, we characterized the structures and cytotoxic activity of the peptides used in this work. The 3D structure of the Tag7 protein (PGLYRP1) and the peptides studied in this work are shown in Figure 1a. The peptides are labeled as follows: 17.0 (RNVQHYHMKT) is marked in green, peptide N9 (PASCQQQAR) is marked in red, and peptide N15 (RYVVVSHTAGSSCNT) is marked in yellow. As shown in our previous study [48], peptide 17.0 blocked the interaction between Tag7 and the TREM-1 receptor. Peptides N9 and N15 are part of the biologically active Tag7 epitope, peptide N3, which induces TREM-1-dependent cytotoxicity of lymphocytes. Therefore, we investigated the regulatory effect of these peptides on the activity of cytotoxic lymphocytes.

The synthesized peptides were incubated with isolated human PBMC for 6 days. Thereafter, the activated lymphocytes were added to MHC-negative tumor cells of the K562 line, and their cytotoxic effect on these cells was determined. The results are shown in Figure 1b. Individual shortened fragments of the N3 peptide do not reproduce the biological activity of this peptide.

It was shown that peptides N9, N15, and 17.0 do not activate cytotoxic lymphocytes. Next, we tested the possibility of binding of these peptides to the TREM-1 receptor in solution and on the cell surface and their activation under the action of these peptides.

### 2.2. Peptides 17.0, N9, and N15 Bind to the TREM-1 Receptor with High Affinity

First, we evaluated the affinity of the interaction of these peptides with the receptor in solution. The interaction between the selected peptides and the receptor was studied using microscale thermophoresis. The affinity of the studied complexes was assessed based on the change in thermodynamic signals obtained by adding various concentrations of peptides to fluorescently labeled TREM-1. The results shown in Figure 2a–c indicate a change in the thermodynamic signal with the change in the concentration of the studied peptides and demonstrate clear binding curves. The calculated dissociation constants have nanomolar values (the Kd of peptide 17.0 is 1.32 ± 0.2 nM; the Kd of peptide N9 is 9.66 ± 0.5 nM; and the Kd of the N15 peptide is 7.43 ± 0.4 nM).

Thus, it can be seen that all three peptides have a high affinity for the TREM-1 receptor. They can form a stable complex with the soluble part of the receptor in solution.

### 2.3. Peptides 17.0, N9, and N15 Activate the TREM-1 Receptor on the Cell Surface

Next, the ability of these peptides to interact with TREM-1 on the cell surface and induce activation of this receptor was investigated. At first, the binding between these peptides and TREM-1 on the cell surface was studied using confocal microscopy (Figure 3). Specific antibodies revealed the presence of TREM-1 on the cell surface of the studied cells. The results shown in this figure also indicate that all three peptides, 17.0, N9, and N15, bind to the cell surface. Colocalization of all three peptides with a receptor can also be seen. The percentage of double-stained cells was determined using the data presented in Appendix A. Based on these results, it can be assumed that these peptides interact with TREM-1 also on the cell surface.

After showing the interaction between the studied peptides and TREM-1 on the cell surface, the ability of these peptides to induce receptor activation was investigated. As mentioned above, upon activation of TREM-1, dimerization of the receptor occurs, followed by dissociation of the soluble exodomain of the receptor, sTREM-1. For this purpose, the appearance of sTREM-1 in a conditioned medium of monocytes incubated with peptides was evaluated using an enzyme immunoassay (Figure 4). It can be seen that peptides 17.0, N9, and N15 induce sTREM-1 dissociation. The release of the exodomain is dose-dependent on the concentration of peptides.

Thus, we demonstrated the binding of all three peptides to TREM-1 on the cell surface, followed by activation of this receptor, which causes sTREM-1 dissociation. However, none of the peptides induced cytotoxic activity of lymphocytes.

### 2.4. Simultaneous Addition of Peptides N9 and N15 to PBMC Induces an Intracellular Activation Signal in Cytotoxic Lymphocytes

Ensuring that the studied peptides interact with the TREM-1 receptor and stimulate its activation, we examined in detail the effect of peptides 17.0, N9, and N15 on the induction of the TREM-1-dependent intracellular signal. The inhibitory and activating activities were investigated. To determine the inhibitory activity, PBMC were preincubated with the studied peptides for 1 h, then the activating peptide N3 was added, and the cytotoxic activity of lymphocytes was determined after 6 days. From the results shown in Figure 5a, it is evident that peptides 17.0 and N9 inhibited the development of cytotoxic activity stimulated by peptide N3. The N15 peptide had no inhibitory effect on the induction of the TREM-1-dependent signal.

To determine the activation of TREM-1 under the action of these peptides, peptides 17.0 and N9, 17.0 and N15, and N9 and N15 were simultaneously added to PBMC, and the cytotoxic activity of lymphocytes was determined after 6 days of incubation (Figure 5b). Surprisingly, the simultaneous addition of peptides N9 and N15 to PMBCs fully reproduces the activity of full-size Tag7 and peptide N3. The combined effect of these peptides stimulated the activation of cytotoxic lymphocytes. However, the simultaneous addition of peptides 17.0 and N15 or 17.0 and N9 did not lead to stimulation of cytotoxic lymphocytes. In addition, the induction of the cytotoxic effect of lymphocytes is TREM-1-dependent. Cytotoxicity of these peptides was inhibited in the presence of antibodies to TREM-1 and its inhibitors, peptides LP17 and 17.0.

These results show that the studied peptides exhibit different functional activities. The results indicate that the inhibitory peptides 17.0 and N9 compete with ligands for binding to the active site of TREM-1. The N15 peptide does not inhibit the interaction of TREM-1 with the ligand; it binds to another site of the receptor. However, the simultaneous addition of N9 and N15 stimulates the activation of cytotoxic lymphocytes. These peptides bind to other sites of the receptor responsible for the generation of an intracellular signal.

### 2.5. Cytotoxic Lymphocytes Stimulated by the Sum of Peptides N9 and N15 Induce Apoptosis and Necroptosis in Tumor Cells

Next, we investigated which subpopulations of cytotoxic lymphocytes are activated by the simultaneous addition of peptides N9 and N15, and which pathways of programmed cell death are induced by these subpopulations. In our previous study, we found that when TREM-1 interacts with ligands, different subpopulations of cytotoxic lymphocytes are activated at different intervals of incubation time. In the present study, we demonstrate that on the fourth day of incubation of the sum of peptides N9 and N15 with PBMCs, NK cells and CD4+T lymphocytes are activated; in comparison, on the sixth day, CD4+ and CD8+T lymphocytes are activated (Appendix A).

NK cells recognize the MicA molecule on target tumor cells using the NKG2D receptor and kill them with the aid of granzymes. A subpopulation of cytotoxic CD8+T lymphocytes also recognizes MicA using the NKG2D receptor; in this case, however, they are eradicated through FasL-Fas interaction. The cytotoxic CD4+T lymphocyte subpopulation recognizes the Hsp70 molecule on tumor cells using Tag7 on its surface and also kills them through FasL-Fas interaction. Tumor cell death is induced through apoptosis involving caspases 3 and 8 or through necroptosis involving RIP kinases (Appendix A).

Thus, just as it was shown for other TREM-1 ligands (Tag7, peptide N3, Hsp70, HMGB1), cytotoxic lymphocytes activated by the sum of peptides N9 and N15 induced apoptosis and necroptosis in tumor cells.

### 2.6. Molecular Docking of N9, N15, and 17.0 Peptides to the TREM-1 Dimer

To search for the receptor sites responsible for interacting with the found peptides, we used the molecular docking method. The three peptides, N9, N15, and 17.0, were docked to the dimer of TREM-1 (PDB 1Q8M). Each obtained complex was further processed for a 200 ns MD run. Based on the RMSD values of backbone atoms (Appendix A), the N15 peptide suffers the greatest shift from the initial positions, whereas N9 and 17.0 have moderate RMSD values along most parts of the trajectories.

Interacting residues were extracted, and interaction fingerprints (IFPs) were also calculated for each peptide and TREM-1. Residues involved in protein–peptide interactions were considered stable and meaningful if they existed in more than 50% of the trajectory. These residues and corresponding interactions are presented in Appendix A, and RMSD values for these interfacial residues are presented in Appendix A. As is evident, the greatest conformational shift was again observed in the interface of the N15–TREM-1 complex.

Based on the RMSDs of the interfacial residues, each trajectory was clustered, and representative conformations were extracted for each complex.

N15–TREM-1 comprises two major clusters, whereby the first cluster describes 69% of conformations and the second one describes 20% of conformations. The representative conformations for these clusters are named conformation 1 and 2, respectively, and are presented in Figure 6.

In both conformations, the N15 peptide has an extended conformation and is located in the groove on the surface of subunit A of TREM-1. The major difference between the two conformations is that in conformation 1, the N15 fully interacts with TREM-1, whereas in conformation 2, residues 1–5 of N15 are exposed to solvent and do not interact with TREM-1. N15 forms the following H-bonds, Ser6–Ile58A, Cys13–Tyr111A, and Asn14–Arg84A, and pi-cation interaction, His7-Arg59A, in both conformations. In conformation 1, the additional salt bridge R1–E30A is observed, and the H-bond between Ser6 and Arg59 is unique to the second conformation.

The 17.0 peptide adopts an extended conformation and lies in a groove on the interface of subunits A and B of TREM-1 (Figure 7a). As seen in Figure 7a and Appendix A, the 17.0 peptide interacts with both subunits of TREM-1 via vdW, hydrophobic, ionic interactions, and H-bonds. The following H-bonds are formed, Arg1–Glu33B, Gln4–Glu106B, Gln4–Ser108B, His5–Ser108B, His5–Leu131B, and Tyr6–Gln117A, in addition to a salt bridge, Arg1–Glu33. From the above results, it can be concluded that for most H-bonds and salt bridges, the peptide forms with subunit B of the TREM-1.

N9 also adopts an extended conformation lying on both subunits A and B of TREM-1 and forming multiple stable interactions with the protein (Figure 7b and Appendix A). In addition to vdW and hydrophobic interaction, N9 forms some specific interactions such as 3 salt bridges (Pro1–Glu106B, Arg9–Arg72B, and Arg9–Glu121A) and 12 hydrogen bonds (Pro1–Glu106B, Pro1–Ser108B, Ala2–Glu106B, Cys4–Arg130B, Gln6–Gln117, Gln7–Arg72, Ala8–Gln52A, Arg9–Lys47A, Arg9–Ser50A, Arg9–Arg72A, and Arg9–Glu121A).

Summarizing the above results, the sites responsible for the interaction of the studied peptides with the TREM-1 receptor were identified.

## 3. Discussion

Two important findings from the present study should be highlighted: (1) the shortened peptide fragments of the Tag7 protein responsible for opposite functions have been identified (inhibition of the interaction of the TREM-1 receptor with ligands and induction of cytotoxic action of lymphocytes); (2) the confirmation of these results using a mathematical model.

In the present study, we identified and characterized peptides of Tag7 epitopes located in the N terminal region of the polypeptide chain responsible for the activation of the TREM-1 receptor. Peptides 17.0, N9, and N15 exhibited different functional activities. Peptide 17.0 is responsible for interaction with the receptor and competes with the ligand by binding to the site of the receptor responsible for contact with the ligand. Peptide N9 exhibits bifunctional activity: similar to peptide 17.0, it also possesses a site responsible for binding to the receptor, but also contains a fragment involved in receptor activation. The N15 peptide, taken separately, exhibits neither inhibitory nor activating activities but is vital, together with the N9 peptide, to reproduce the activity of full-size Tag7. From the above results, it can be assumed that the TREM-1 receptor, in addition to the TNFR1 receptor, possesses two epitopes that induce the activation of TREM-1 after simultaneous interaction with peptides N9 and N15.

Molecular modeling confirmed the above results. We conducted MD simulations for the docking-derived complexes of peptides N15, 17.0, and N9 with the TREM-1 protein and calculated IFPs for each one. Based on these findings, we concluded that the N15 peptide adopts two major conformations in complex with TREM-1, where its first five residues are bound to the protein surface or exposed to the solvent. We hypothesize that the removal of these first five residues may improve the stability of the N15–TREM-1 complex. This supposed reduced peptide could form up to five highly specific H-bonds, one pi-cation interaction, and multiple non-specific hydrophobic and vdW interactions.

Regarding the 17.0 peptide, it forms five H-bonds and one salt bridge with TREM-1 through Arg1, Gln4, His5, and Tyr6. Met8 only participates in hydrophobic interactions, with the other residues not forming any stable interactions. From these results, we were able to conclude that the first six residues of the 17.0 peptide may serve as the basis for the development of strong and highly specific TREM1 binders.

Despite having the shortest length, the N9 peptide binds TREM-1 with the largest number of specific bonds (12 H-bonds and 3 salt bridges), making it the most likely candidate for the development of peptidomimetics or other low-molecular-weight TREM-1 binders. An in-depth study examining N9–TREM-1 interactions may provide useful pharmacophore models that could be used for the virtual screening of chemical databases.

We also found TREM-1 residues that participate in the binding of at least two peptides, with these residues including Ile57, Ile58, Glu106, Ser108, Leu110, Tyr116, Pro118, Pro119, Arg130, and Val 132. The majority of these residues participate in hydrophobic and vdW interactions, with only two of them (Ile58 and Glu106) forming H-bonds. We hypothesize that these residues are essential for the binding of peptides with TREM-1 and that other residues participate in H-bonds, salt bridges, and pi-cation interactions, providing specificity. The residues found may represent key focuses in the development of TREM-1–binders via de novo design or virtual screening.

The results of computer modeling showed that all three peptides have a common binding site, interacting with amino acid residues 105–111 of the TREM-1 molecule. However, each of the studied peptides has unique binding sites, which, from the above evidence, determine the differences in their functional activity. The region from 116 to 121 amino acids interacts with peptides 17.0 and N9. Both of these peptides, when added to the TREM-1 receptor, cause the receptor to block and prevent its subsequent activation. It is possible that this region, 116–121, is responsible for the inhibitory effect. The second interesting feature of the interaction between peptides 17.0 and N9 and the TREM-1 receptor is the simultaneous interaction with two receptor molecules. This finding suggests that the binding of peptides 17.0 and N9 leads to receptor dimerization. The N9 peptide has a unique site of interaction with TREM-1, which is not present in the other peptides studied. The region between amino acids 47 and 52 is likely one of the two points required for receptor activation when the N9 peptide is added together with N15, and this sum of peptides activates the receptor. When studying the interaction between N15 and TREM-1, it is immediately apparent that N15 interacts with only one molecule of TREM-1. The N15 peptide possesses several unique binding sites for the TREM-1 molecule. There are two unique regions of the TREM-1 molecule with which N15 interacts. Of these regions, sites 28–30 and 58–59 are the most suitable, the interaction with which is necessary, together with sites 47–52, for receptor activation.

Based on the above data, it can be assumed that there are two stages of ligand–receptor interaction. During the first stage, the ligand binds in the active center of TREM-1; during the second stage, the ligand bound to the receptor interacts with two active epitopes. For further study of the anticancer effect of Tag7 peptides, in vivo experiments are required. Activation of the antitumor lymphocytes via Tag7 peptides can be combined with the anticheckpoint therapy to further enhance its anticancer activity [50,51,52].

## 4. Materials and Methods

### 4.1. Cell Cultivation and Sorting

K562, RAW264.7 cells were cultivated in RPMI-1640 (Himedia Laboratories Private Limited, Maharashtra, India) with 2 mM L glutamine, 10% FCS (Invitrogen, Carlsbad, CA, USA), with 1% kanamycin (Thermo Fisher Scientific, Waltham, MA, USA). Human peripheral blood mononuclear cells (PBMC) were isolated from the total leukocyte pool of healthy donors via centrifugation using a Ficoll–Paque density gradient (Cytiva Livescience^TM^, Marlborough, MA, USA) and cultured at a density of 4 × 10^6^ cells/mL in RPMI-1640 (Himedia Laboratories Private Limited, Maharashtra, India) with Tag7, peptides N3, N9, N15, or 17.0 (all peptides were added at a concentration of 10^−9^ M for 6 days. All procedures performed were in accordance with the Declaration of Helsinki (1964) and its later amendments (World Medical Association, 2013) or comparable ethical standards and were approved by the medical ethics committee of FSBI N.N. Blokhin National Medical Research Center of Oncology of the Ministry of Health of the Russian Federation. CD4+ (CD8+) T-lymphocytes were isolated from the PBMCs using the Dynabeads^TM^ Untouched^TM^ Human CD4 T Cells Kit, and NK cells were isolated from PBMC by Dynabeads^TM^ Untouched^TM^ Human NK Cells Kit according to the manufacturer’s protocol (Thermo Fisher Scientific, Waltham, MA, USA).

### 4.2. Proteins and Antibodies

Recombinant Tag7 and sTREM-1 were obtained using the method described in [42].

N9 (PASCQQQAR), N15 (RYVVVSHTAGSSCNT), and 17.0 (RNVQHYHMKT) peptides were synthesized based on the method described in [47].

The inhibitory peptides LP17 (LQVTDSGLYRCVIYHPP), N9, N15, and 17.0 at concentrations of 10^−9^ M were pre-incubated with lymphocytes for 1 h before the addition of Tag7.

### 4.3. Cytotoxicity Assays

For cytotoxicity tests, cells of the K562 line were cultured in 96-well plates (Luoyang Fudau Biotech Co, Luoyang, Henan, China) (6 × 10^4^ per well) before the addition of lymphocytes (1.2 × 10^6^ per well) and then incubated at 37 °C in an atmosphere with 5% CO2 for 3–20 h. Cytotoxicity was measured using a Cytotox 96 analysis kit (Promega, Madison, WI, USA) after 24 h of incubation in accordance with the manufacturer’s protocol. In the inhibition assays, cells were pre-incubated for 1 h with the 17.0, N9, N15, Caspase 3 inhibitor Ac-DEVD-CHO (5 μM), caspase 8 inhibitor Ac-IEID-CHO (5 μM), necrostatin 1 (5 μM), and RIP1 kinase inhibitor (all from Thermo Fisher Scientific, Waltham, MA, USA). Polyclonal antibodies to Tag7 were obtained from ABclonal (ABclonal, Woburn, MA, USA). Monoclonal antibodies to TREM-1 were obtained from Thermo Fisher Scientific (Thermo Fisher Scientific, Waltham, MA, USA).

### 4.4. IFA

The level of sTREM-1 secretion was tested using a Human TREM-1 ELISA kit (Thermo Fisher Scientific, Waltham, MA, USA) in accordance with the manufacturer’s protocols.

### 4.5. Microscale Thermophoresis

The purified sTREM-1 protein was labeled using an Alexa Fluor^TM^ 633 Protein Labeling Kit (Life Technologies Corporation, Eugene, OR, USA) in accordance with the manufacturer’s protocol [53].

The 17.0, N9, and N15 peptides (C = 200 nM for all) were incubated for 30 min in the dark at room temperature in 16 different concentrations obtained through sequential dilution, starting with the highest soluble concentration. The samples were transferred to glass capillaries (Monolith NT Capillaries) and analyzed via microscale thermophoresis using a Nano-Temperature Monolith NT 115 device (24% IR laser power). The signal quality was monitored using a NanoTemper Monolith device to detect possible autofluorescence of the ligand, aggregation, or changes in the photobleaching rate. The experiments were carried out in at least three replicates and processed using affinity analysis software (MO Control v.1.6.1, NanoTemper Technologies GmbH, München, Germany).

### 4.6. Confocal Microscopy

RAW264.7 cells were grown on glass coverslips, and peptides 17.0, N9, and N15 were added. Then, fixed with 4% formaldehyde. The cells were rinsed three times with PBS, and then the samples were placed into blocking solution (1% BSA in PBS) for 30 min at room temperature. The TREM-1 receptor was stained with monoclonal mouse antibodies against TREM-1 and a Goat anti-mouse IgG (H+L) Cross-Adsorbed Secondary Antibody, Alexa Fluor^TM^ 488 (Molecular Probes By Life Technologies, Carlsbad, CA, USA). The 17.0 (N9 and N15) peptide was stained with polyclonal rabbit antibodies against Tag7 and a Goat anti-Rabbit IgG (H+L) Cross-Adsorbed Secondary Antibody, Alexa Fluor^TM^ 633. Following washing with PBS, the coverslips were embedded in ProLong Gold (Thermo Fisher Scientific, Waltham, MA, USA). Fluorescence images were obtained using a Leica STELLARIS 5 confocal microscope (Leica, Wetzlar, Germany) and then analyzed using Leica confocal software (2.61.15) and processed in ImageJ 1.54f (LOCI, Madison, WI, USA).

### 4.7. Statistical Analysis

All of the data presented in the article were the result of at least three replicates. The data are presented as the average ± standard deviation. Testing for significant differences between the treatment and control was performed using MathCad software (version 15.0, PTC, Cambridge, Great Britain). In experiments on the treatment of cells with one agent, Student’s criteria were used. For experiments on the treatment of cells with two or more agents, two-factor analysis of variance was used. The results are presented as average values ± SDs. The value of *p* < 0.05 was considered statistically significant. GraphPad Prism 8.0 software was used for data presentation.

### 4.8. Molecular Docking

The atomic model of the TREM-1 dimer was downloaded from the Protein Data Bank (PDB ID: 1Q8M). This model served as a target for peptide–protein docking with AutoDock CrankPep v.1.0. The docking calculations were initiated as a mixture of coiled and helical conformations for N15 (RYVVVSHTAGSSCNT), 17.0 (RNVQHYHMKT), and N9 (PASCQQQAR) peptides. Calculations for each peptide were conducted in 500 replicas. Each replica contained 3 × 10^6^ steps per peptide monomer. Thereafter, the docked peptides were selected based on their protein–peptide interaction profiles.

All three complexes obtained through docking served as starting points for molecular dynamics (MD) simulations in the GROMACS 2021.4 software. Each complex was solvated with TIP3P water in the presence of 0.150 M NaCl and minimized with 50,000 steps. Thereafter, all complexes were equilibrated in NVT and NPT ensembles for 2 ns and 4 ns, respectively. The movements of protein and peptide atoms were restricted during the equilibration phase. The equilibrated complexes were used for a production MD run for 200 ns. A V-rescale thermostat and Parrinello–Rahman barostat were used during this procedure. The temperature and pressure were maintained at 298 K and 1 bar, respectively. The timestep was 2 fs. The cutoff for calculation of non-bonded interactions and short-range electrostatics was set to 12 Å. Long-range electrostatics was calculated with the particle mesh Ewald (PME) method.

All trajectories were analyzed to determine intermolecular interactions between the peptides and TREM-1 with the ProLIF Python 3.9 library. The following interactions were considered during this study: hydrophobic, van der Waals (vdW), hydrogen bonds (H-bonds), salt bridges, and pi-cations.

## 5. Conclusions

In this study, we were able to identify several peptides of the innate response protein PGLYRP1 capable of regulating the activity of the TREM-1 receptor on monocytes, which triggers the activation of antitumor lymphocytes. The conducted molecular docking experiments revealed amino acid residues necessary for various functions of peptides, thus providing a basis for the development of therapeutic drugs to regulate the receptor’s activity in the treatment of autoimmune diseases and tumors.

## Figures and Tables

**Figure 1 ijms-26-04069-f001:**
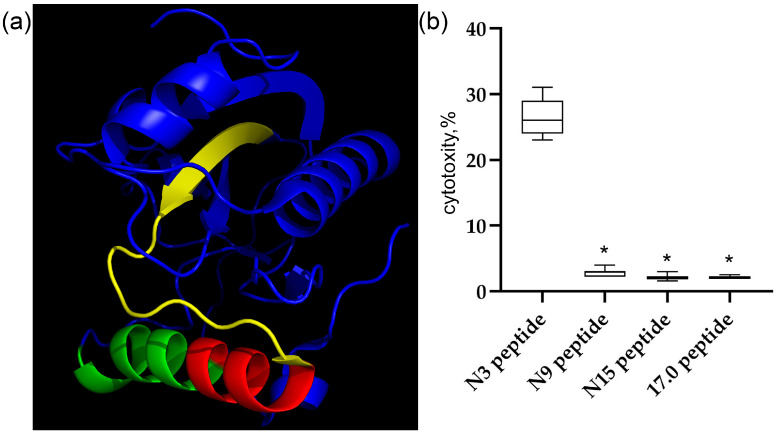
The Tag7 peptides do not activate the TREM-1 receptor on human monocytes. (**a**) The structure of the Tag7 protein. Peptide 17.0 (RNVQHYHMKT) is marked in green, peptide N9 (PASCQQQAR) is marked in red, and peptide N15 (RYVVVSHTAGSSCNT) is marked in yellow. (**b**) The cytotoxic activity of human PBMC incubated for 6 days with Tag7 peptides, tested on K562 cells. *n* = 5 for each point (* *p*-value < 0.05).

**Figure 2 ijms-26-04069-f002:**
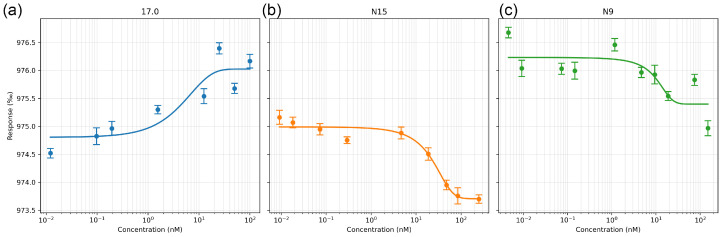
Microscale thermophoresis data for the interaction between sTREM-1 and Tag7 peptides: (**a**) sTREM-1 with the 17.0 peptide, (**b**) sTREM-1 with the N15 peptide, and (**c**) sTREM-1 with the N9 peptide. Each experiment was carried out in triplicate, with a selection of the most common data shown.

**Figure 3 ijms-26-04069-f003:**
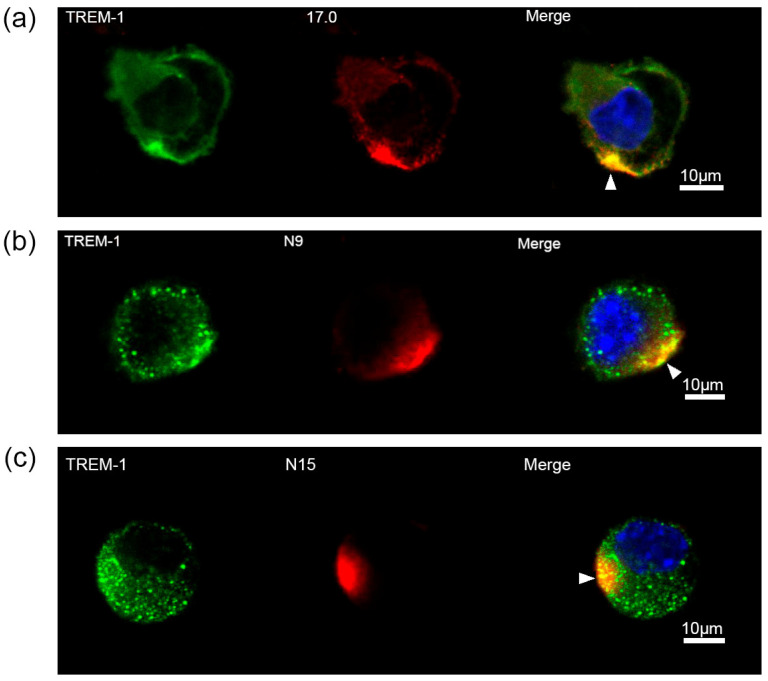
Tag7 peptides bind to the surface of RAW264.7 cells and have colocalization with the TREM-1 receptor. (**a**) Confocal micrograph of the 17.0 peptide (red) and TREM-1 (green) and layer superposition (yellow) on the surface of RAW264.7 cells. Roughly 69% of all observed cells were double-stained (*n* = 4). (**b**) Confocal micrograph of the N9 peptide (red) and TREM-1 (green) and layer superposition on the surface of RAW264.7 cells. Roughly 71% of all observed cells were double-stained (*n* = 4). (**c**) Confocal micrograph of the N15 peptide (red) and TREM-1 (green) and layer superposition on the surface of RAW264.7 cells. Roughly 68% of all observed cells were double-stained (*n* = 4). Arrows indicate the colocalization region. DAPI staining is blue.

**Figure 4 ijms-26-04069-f004:**
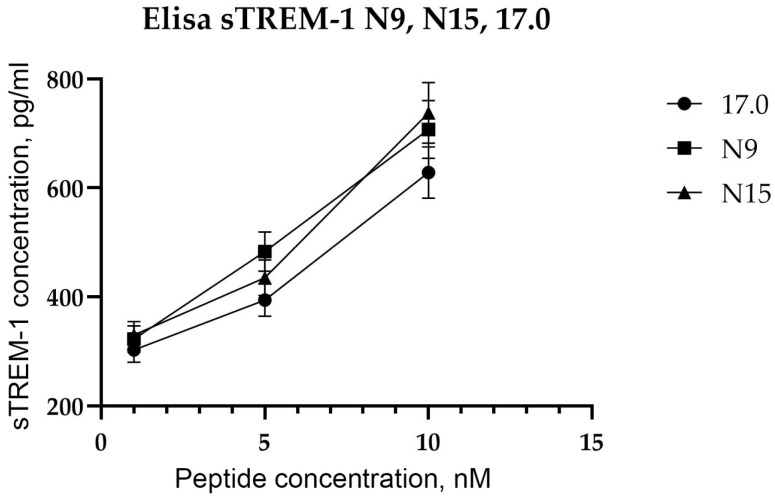
Tag7 peptides activate the TREM-1 receptor on human monocytes. ELISA analysis of soluble sTREM-1 released in the solution by human monocytes from PBMCs treated with several different concentrations of 17.0, N9, and N15 peptides.

**Figure 5 ijms-26-04069-f005:**
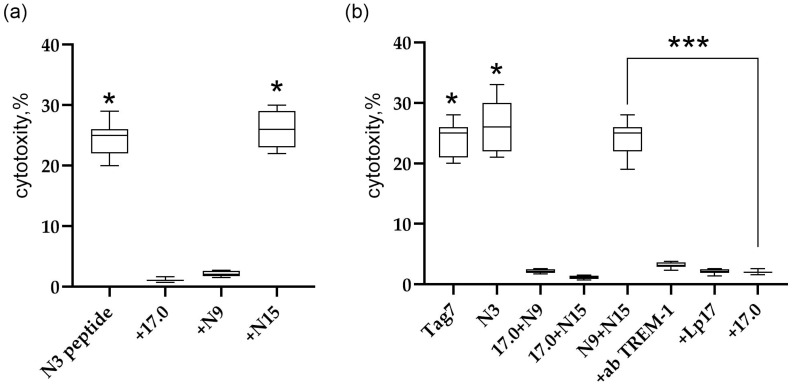
Regulation of TREM-1 activity via Tag7 peptides. (**a**) Cytotoxic activity of PBMC treated with the N3 peptide alone or 1 h after the addition of 17.0, N9, or N15. (**b**) Cytotoxic activity of PBMC treated with Tag7, the N3 peptide, and the sum of 17.0 and N9 peptides, 17.0 and N15 peptides, and N9 and N15 peptides. The cytotoxicity of the sum of N9 and N15 peptides in the presence of antibodies to TREM-1, the Lp17 peptide, and the 17.0 peptide. * *p* < 0.05, *** all results *p* < 0.05.

**Figure 6 ijms-26-04069-f006:**
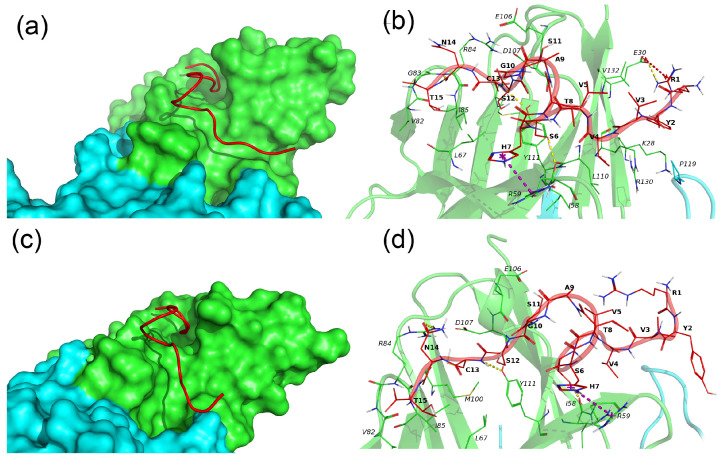
Binding of N15 peptide with the TREM-1 dimer. (**a**–**c**) Binding modes of N15 with TREM-1 (PDB 1Q8M) in conformation 1 and 2, correspondingly. The N15 peptide is shown as a red computer-generated image, TREM-1 is shown as a surface representation, subunit A is marked in green, and subunit B is marked cyan. (**b**–**d**) Intermolecular interactions between N15 and TREM-1. The N15 peptide is marked in red, and subunits A and B of TREM-1 are marked in green and cyan, respectively. All residues of N15 and interacting residues of TREM-1 are shown as lines; H-bonds are marked by yellow dashed lines, and pi-cation interactions are marked by magenta dashed lines. Hydrophobic and vdW interactions are not shown. One-letter naming of residues is used herein, with the peptide residues shown in bold and the protein residues shown in italics.

**Figure 7 ijms-26-04069-f007:**
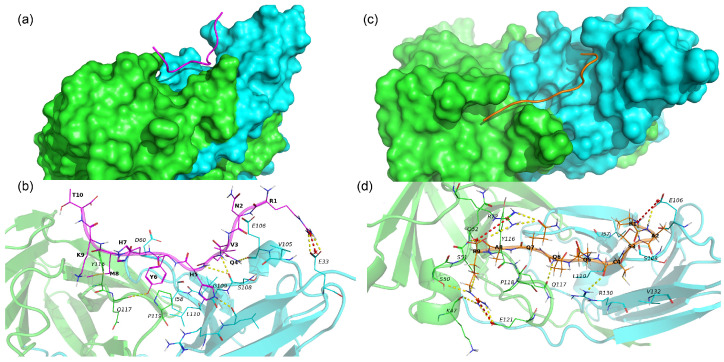
Binding of 17.0 and N9 peptides with TREM-1 dimer. (**a**) Binding mode of peptide 17.0 with TREM-1 (PDB 1Q8M). The 17.0 peptide is shown as a red computer-generated image, and TREM-1 is shown as a surface representation. Subunit A is marked in green, and subunit B is marked in cyan. (**b**) Intermolecular interactions between peptide 17.0 and TREM-1. The 17.0 peptide is marked in purple, and subunits A and B of TREM-1 are marked in green and cyan, respectively. All residues of 17.0 and interacting residues of TREM-1 are shown as lines. H-bonds are shown as yellow dashed lines, and salt bridges are shown as red dashed lines. Hydrophobic and vdW interactions are not shown. One-letter naming of residues is used herein, with the peptide residues shown in bold and the protein residues shown in italics. (**c**) Binding mode of the N9 peptide with TREM-1. The N9 peptide is shown as a golden computer-generated image, and TREM-1 is shown as a surface representation. Subunit A is marked in green, and subunit B is marked in cyan. (**d**) Intermolecular interactions between N9 and TREM-1. The N9 peptide is marked in gold, and subunits A and B of TREM-1 are marked in green and cyan, respectively. All residues of the N9 peptide and interacting residues of TREM-1 are shown as lines. H-bonds are shown as yellow dashed lines, and salt bridges are shown as red dashed lines. Hydrophobic and vdW interactions are not shown. One-letter naming of residues is used herein, with the peptide residues shown in bold and the protein residues shown in italics.

## Data Availability

Data are contained within the article and Appendix A.

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
