# Peer review of "Shortened PGLYRP1 Peptides Regulate Antitumor Activity of Cytotoxic Lymphocytes via TREM-1 Receptor: From Biology to Bioinformatics"

_ijms, 2025, doi:10.3390/ijms26094069_

Round 1

Reviewer 1 Report

Comments and Suggestions for Authors

Type: Article

Title: PGLYRP1 short peptides regulate TREM-1 induced anti-tumor activity of cytotoxic lymphocytes.

Authors: Daria M. Yurkina , Kirill A. Shcherbakov , Elena A. Romanova , Anna V. Tvorogova , Alexey M. Feoktistov , Georgii P. Georgiev , Denis V. Yashin * , Lidia P. Sashchenko

Section: Molecular Biophysics

Comments to the Author,

In the present work, the authors studied how the pro-inflammatory immune response is crucial for defending the body against pathogens and tumors. The author studied some peptides derived from the innate immune protein PGLYRP1 that regulate the activity of the TREM-1 receptor. This receptor, expressed on monocytes, plays a key role in activating antitumor lymphocytes. The authors used molecular docking analysis, providing valuable insights into the receptor's activation mechanism.

The manuscript has very interesting results. However, it is not well written and in some parts is not clear. The paper requires some important revision before being accepted for publication.

  • I suggested to organize the introduction, the mini paragraphs look as loose ideas, not like a cohesive text. Including the last paragraph of the introduction that is describe as a list of activities must be changed.
  • Line 107 looks extremely similar to line 113, including that both are in italics
  • Figure 2 must be change including the statistics and the standard deviation of each point. Other wise is not valid
  • Figures 3,4,5 are not in Palatino linotype fond, they must be changed
  • Figure 6 is in Russian and is not in Palatino linotype fond. The size must be reduced
  • Figure 7, 8 and 9 The size must be reduced and the Palatino linotype fond is missibg

The work needs important changes for being suitable for publication in International Journal of Molecular Sciences

Minor changes

  • I suggest to reduce the size of the figure to make them look nicer. Because if you see the size of the letters inside the graphs is too big
  • Line 130 and 131 the symbol ± must has space before and after the numbers
  • Line 140, the way the figure 3 is cited in the text is different to figure 1
  • Reduce the size of figure 4 and 5 and used the Paltino linotype fond according to the template
Comments on the Quality of English Language

The work is very interesting and the potential applications of the peptides in the pro inflammatory immune response are very important. However, the way the authors write is affecting the very interesting results. I suggest and important edition of the manuscript, maybe with the help of a native speaker. Otherwise is impossible to be according to the standards of the journal.

Author Response

Open Review

Quality of English Language

(x) The English could be improved to more clearly express the research.
( ) The English is fine and does not require any improvement.

Yes

Can be improved

Must be improved

Not applicable

Does the introduction provide sufficient background and include all relevant references?

( )

( )

(x)

( )

Is the research design appropriate?

( )

( )

(x)

( )

Are the methods adequately described?

( )

(x)

( )

( )

Are the results clearly presented?

( )

( )

(x)

( )

Are the conclusions supported by the results?

( )

( )

(x)

( )

Comments and Suggestions for Authors

Type: Article

Title: PGLYRP1 short peptides regulate TREM-1 induced anti-tumor activity of cytotoxic lymphocytes.

Authors: Daria M. Yurkina , Kirill A. Shcherbakov , Elena A. Romanova , Anna V. Tvorogova , Alexey M. Feoktistov , Georgii P. Georgiev , Denis V. Yashin * , Lidia P. Sashchenko

Section: Molecular Biophysics

Comments to the Author,

In the present work, the authors studied how the pro-inflammatory immune response is crucial for defending the body against pathogens and tumors. The author studied some peptides derived from the innate immune protein PGLYRP1 that regulate the activity of the TREM-1 receptor. This receptor, expressed on monocytes, plays a key role in activating antitumor lymphocytes. The authors used molecular docking analysis, providing valuable insights into the receptor's activation mechanism.

The manuscript has very interesting results. However, it is not well written and in some parts is not clear. The paper requires some important revision before being accepted for publication.

  • I suggested to organize the introduction, the mini paragraphs look as loose ideas, not like a cohesive text. Including the last paragraph of the introduction that is describe as a list of activities must be changed.

We thank the reviewer for useful comments and the interest to our work. We have changed the Introduction Section to address this comment.

  • Line 107 looks extremely similar to line 113, including that both are in italics

We have changed this.

  • Figure 2 must be change including the statistics and the standard deviation of each point. Other wise is not valid

We have reworked Figure 2 to address this point.

  • Figures 3,4,5 are not in Palatino linotype fond, they must be changed

We have changed figures 3-5 to address this issue.

  • Figure 6 is in Russian and is not in Palatino linotype fond. The size must be reduced

We apologize for our mistake, The figure is changed.

  • Figure 7, 8 and 9 The size must be reduced and the Palatino linotype fond is missibg

We have changed figures 7-9 to address this.

The work needs important changes for being suitable for publication in International Journal of Molecular Sciences

Minor changes

  • I suggest to reduce the size of the figure to make them look nicer. Because if you see the size of the letters inside the graphs is too big

done

  • Line 130 and 131 the symbol ± must has space before and after the numbers

done

  • Line 140, the way the figure 3 is cited in the text is different to figure 1

done

  • Reduce the size of figure 4 and 5 and used the Paltino linotype fond according to the template

done

Comments on the Quality of English Language

The work is very interesting and the potential applications of the peptides in the pro inflammatory immune response are very important. However, the way the authors write is affecting the very interesting results. I suggest and important edition of the manuscript, maybe with the help of a native speaker. Otherwise is impossible to be according to the standards of the journal.

Submission Date

11 March 2025

Date of this review

19 Mar 2025 12:09:49

Reviewer 2 Report

Comments and Suggestions for Authors

The pro-inflammatory immune response is critical to protect the body from tumors. This article identified several peptides derived from the innate response protein PGLYRP1, which could regulate the activity of the TREM-1 receptor on monocytes and trigger the activation of antitumor lymphocytes. Moreover, molecular docking provided valuable references for understanding the activation mechanism of TREM-1 receptor. Collectively, this article should be of interest to readers of Int. J. Mol. Sci. and I would recommend the publication of this article after addressing minor corrections. 

minor comments/suggestions

  1. Page 3.To help readers and potential users, please outline the reason why peptides 17.0, N9 and N15 were selected in this manuscript.
  2. Page 4, line 131.Please change “nm” to “nM”.
  3. Page 9, Figure 7.The PDB number of proteins should be provided in the caption of figures.
  4. As the in vitro experiments are not sufficient to verify the anticancer potential of PGLYRP1-derived peptides, further in vivo experiments are needed in the future study. The authors should point out this issue for readers.
  5. Except the pro-inflammatory immune response, "immune checkpoint blockade" (eg, blockade of PD-1/PD-L1 interactions) and "oncolytic" (Oncolytic immunotherapy) effects could also activate the immune system, significantly enhancing the anticancer effect of cytotoxic lymphocytes. Over the last two decades, immune checkpoint blocking peptides and oncolytic peptides have emerged as striking agents to combat cancers, especially drug-resistant refractory malignancies. To help readers and potential users, it would be appropriate to cite the representative work on the development of checkpoint blocking peptides (suggest, Chem, 2024, 10, 2390.) and oncolytic peptides (suggest, J. Med. Chem., 2024, 67, 3885. Acta Pharmacol. Sin., 2023, 44, 201. Bioorg. Chem., 2023, 138, 106674.).
  6. It is recommended to have a native English-speaking professional polish the manuscript.
Comments on the Quality of English Language

It is recommended to have a native English-speaking professional polish the manuscript.

Author Response

Open Review

Quality of English Language

(x) The English could be improved to more clearly express the research.
( ) The English is fine and does not require any improvement.

Yes

Can be improved

Must be improved

Not applicable

Does the introduction provide sufficient background and include all relevant references?

( )

(x)

( )

( )

Is the research design appropriate?

(x)

( )

( )

( )

Are the methods adequately described?

(x)

( )

( )

( )

Are the results clearly presented?

(x)

( )

( )

( )

Are the conclusions supported by the results?

(x)

( )

( )

( )

Comments and Suggestions for Authors

The pro-inflammatory immune response is critical to protect the body from tumors. This article identified several peptides derived from the innate response protein PGLYRP1, which could regulate the activity of the TREM-1 receptor on monocytes and trigger the activation of antitumor lymphocytes. Moreover, molecular docking provided valuable references for understanding the activation mechanism of TREM-1 receptor. Collectively, this article should be of interest to readers of Int. J. Mol. Sci. and I would recommend the publication of this article after addressing minor corrections. 

minor comments/suggestions

  1. Page 3.To help readers and potential users, please outline the reason why peptides 17.0, N9 and N15 were selected in this manuscript.

We thank the reviewer for the interst to our work and useful comments. We have added information in the Introduction Section to address this comment.

  1. Page 4, line 131.Please change “nm” to “nM”.

done

  1. Page 9, Figure 7.The PDB number of proteins should be provided in the caption of figures.

Added.

  1. As the in vitro experiments are not sufficient to verify the anticancer potential of PGLYRP1-derived peptides, further in vivo experiments are needed in the future study. The authors should point out this issue for readers.

We have added this point to the end of Discussion Section.

  1. Except the pro-inflammatory immune response, "immune checkpoint blockade" (eg, blockade of PD-1/PD-L1 interactions) and "oncolytic" (Oncolytic immunotherapy) effects could also activate the immune system, significantly enhancing the anticancer effect of cytotoxic lymphocytes. Over the last two decades, immune checkpoint blocking peptides and oncolytic peptides have emerged as striking agents to combat cancers, especially drug-resistant refractory malignancies. To help readers and potential users, it would be appropriate to cite the representative work on the development of checkpoint blocking peptides (suggest, Chem, 2024, 10, 2390.) and oncolytic peptides (suggest, J. Med. Chem., 2024, 67, 3885. Acta Pharmacol. Sin., 2023, 44, 201. Bioorg. Chem., 2023, 138, 106674.).

We have added this information to the Discussion Section.

  1. It is recommended to have a native English-speaking professional polish the manuscript.

We have conducted language editing service on the mdpi site to improve the English language of the manuscript.

Comments on the Quality of English Language

It is recommended to have a native English-speaking professional polish the manuscript.

Submission Date

11 March 2025

Date of this review

17 Mar 2025 09:46:41

Reviewer 3 Report

Comments and Suggestions for Authors

By identifying peptides of the innate response protein PGLYRP1 that regulate the activity of the TREM-1 receptor—known to trigger antitumor lymphocyte activation—the research provides valuable insights into the mechanisms of receptor activation. The findings, including the identification of peptides with inhibitory and activating effects, and molecular docking that highlights essential amino acid residues, have significant implications for the development of therapeutic drugs targeting autoimmune and tumor diseases.However, the manuscript's poor writing quality and lack of adherence to scientific standards severely undermine its potential impact. Extensive revisions are necessary to improve clarity, structure, and data presentation. Only after these improvements can the technical aspects and broader contributions of the study be accurately assessed.

  1. The abstract section misses quantitative data and clarity and must be rewritten.
  2. The introduction is clear and must be focused on the main message of the manuscript. Short paragraphs without a clear message must be avoided.
  3. The description of the results is not clear and does not meet scientific standards.
  4. The figure legends miss general titles before their description.
  5. Introductory and connective sentences are crucial in the results section to enhance clarity and facilitate better understanding of the findings.
  6. The quality of figure 1a can be improved.
  7. Quality of figure 2 can be improved. How many times were this experiment repeated?
  8. Figure 3 seems to present unspecific binding. It is not clear the conclusion made by the authors.
  9. The molecular docking can be summarised in one figure, with additional information presented as supplementary material.
  10. The discussion is not aligned with the introduction and results. It can be shortened with clear logic presentation of the findings.
  11. How was the cytotoxicity % estimated? The authors have not mentioned the negative and positive control used to normalise the data.
  12. The methods sections contain one-sentence paragraphs that can be connected with paragraphs with similar ideas or expanded.
  13. The title does not reflect its content. 

Comments on the Quality of English Language

The English needs revision. 

Author Response

Open Review

Quality of English Language

(x) The English could be improved to more clearly express the research.
( ) The English is fine and does not require any improvement.

Yes

Can be improved

Must be improved

Not applicable

Does the introduction provide sufficient background and include all relevant references?

( )

( )

(x)

( )

Is the research design appropriate?

( )

( )

(x)

( )

Are the methods adequately described?

( )

( )

(x)

( )

Are the results clearly presented?

( )

( )

(x)

( )

Are the conclusions supported by the results?

( )

( )

(x)

( )

Comments and Suggestions for Authors

By identifying peptides of the innate response protein PGLYRP1 that regulate the activity of the TREM-1 receptor—known to trigger antitumor lymphocyte activation—the research provides valuable insights into the mechanisms of receptor activation. The findings, including the identification of peptides with inhibitory and activating effects, and molecular docking that highlights essential amino acid residues, have significant implications for the development of therapeutic drugs targeting autoimmune and tumor diseases.However, the manuscript's poor writing quality and lack of adherence to scientific standards severely undermine its potential impact. Extensive revisions are necessary to improve clarity, structure, and data presentation. Only after these improvements can the technical aspects and broader contributions of the study be accurately assessed.

  1. The abstract section misses quantitative data and clarity and must be rewritten.

We are thankful to the reviewer for careful reading of our work and useful comments. We have changed the Abstract of our work to address this comment.

  1. The introduction is clear and must be focused on the main message of the manuscript. Short paragraphs without a clear message must be avoided.

We have made corrections to the Introduction Section to make it more focused on our main goal and decreased the number of short paragraphs.

  1. The description of the results is not clear and does not meet scientific standards.

We have made corrections to the Results Section to address this comment.

  1. The figure legends miss general titles before their description.

We are thankful to the reviewer for uncovering of our mistake. We have added general titles to all figures.

  1. Introductory and connective sentences are crucial in the results section to enhance clarity and facilitate better understanding of the findings.

We have added several introductory and connective sentences to the Results Section to make it clearer for the potential readers.

  1. The quality of figure 1a can be improved.

Done.

  1. Quality of figure 2 can be improved. How many times were this experiment repeated?

We have reworked Figure 2 to address this comment. The experiment was conducted three times and the most common data is shown on the Figure 2.

  1. Figure 3 seems to present unspecific binding. It is not clear the conclusion made by the authors.

We have done several new experiments and provide more convincing results in the new Figure 3.

  1. The molecular docking can be summarised in one figure, with additional information presented as supplementary material.

We have transferred former Figure 6 to the supplemental and combined former figures 8 and 9 into the one new Figure 7.

  1. The discussion is not aligned with the introduction and results. It can be shortened with clear logic presentation of the findings.

We have shortened the discussion and reworked it to address this comment.

  1. How was the cytotoxicity % estimated? The authors have not mentioned the negative and positive control used to normalise the data.

The method of cytotoxity testing was added to the beginning of the Results Section to clarify this issue. Cytotoxicity was measured using a Cytotox 96 analysis kit (Promega, Madison, WI, USA) according to the manufacturer protocol. It includes several positive and negative controls for each measured point.

  1. The methods sections contain one-sentence paragraphs that can be connected with paragraphs with similar ideas or expanded.

We have improved the Methods Section to address this point.

  1. The title does not reflect its content.

We have changed the Title of the article to make it more connected with the content of the manuscript.

Comments on the Quality of English Language

The English needs revision. 

Submission Date

11 March 2025

Date of this review

22 Mar 2025 21:56:58

Round 2

Reviewer 1 Report

Comments and Suggestions for Authors

They did a good work

Author Response

We thank the reviewer for useful comments.

Reviewer 2 Report

Comments and Suggestions for Authors

This manuscript could be accepted.

Author Response

(The authors gave the same response as above.)

Reviewer 3 Report

Comments and Suggestions for Authors

I have not observed sufficient effort by the authors to address the concerns raised. The manuscript fails to meet the criteria for consideration for publication, as the writing lacks clarity and the intended messages are not effectively conveyed. the quality of figures remains poor. 

Author Response

I have not observed sufficient effort by the authors to address the concerns raised. The manuscript fails to meet the criteria for consideration for publication, as the writing lacks clarity and the intended messages are not effectively conveyed. the quality of figures remains poor.

We have made new work to improve the quality of our manuscript and have made significant changes.

We have changed the Title of the article to reflect bioinformatics section of our work in the Title.

We have shortened Introduction Section by a third, removing all information that is not directly connected to the described data. We have restructured it to make it clearer for the readers and to exclude short paragraphs. We have also added some new information to improve it and make it clearer for the readers.

We have made corrections to the Results Section. We have added introductory and summarizing text to each of the result subsections to provide information about our goals and data obtained. We have also added methods description and made general improvements to make this Section clearer.

We have made excessive changes to the Discussion Chapter. We have shortened this part of the manuscript by a third removing our suggestions about general mechanisms of proinflammatory receptors activation and other material not directly connected with the main goal of our work. We have added some material to make better connection with obtained results. We have restructured it to remove short paragraphs and make it clearer for the readers. We have also improved Methods Section to make it clearer and to remove short paragraphs.

We have reworked all Figures, now they have good resolution, and they take into account all the remarks of all reviewers and fit the requirements of MDPI Authors Instructions.

We hope that these changes will improve the quality of our work.

  1. The abstract section misses quantitative data and clarity and must be rewritten.

We are thankful to the reviewer for careful reading of our work and useful comments. We have changed the Abstract of our work to address this comment. We have added quantitative data and restructured the Abstract to improve it clarity.

  1. The introduction is clear and must be focused on the main message of the manuscript. Short paragraphs without a clear message must be avoided.

We have made excessive corrections to the Introduction Section to make it more focused on our main goal and decreased the number of short paragraphs. We have shortened this part of the manuscript by a third, removing all information that is not directly connected to the described data. We have restructured it to make it clearer for the readers and to exclude short paragraphs. We have also added some new information to improve it.

  1. The description of the results is not clear and does not meet scientific standards.

We have made corrections to the Results Section to address this comment. We have added introductory and summarizing text to each of the result subsections to provide information about our goals and data obtained.

  1. The figure legends miss general titles before their description.

We are thankful to the reviewer for uncovering of our mistake. We have added general titles to all figures.

  1. Introductory and connective sentences are crucial in the results section to enhance clarity and facilitate better understanding of the findings.

We have added several introductory and connective sentences to the Results Section to make it clearer for the potential readers.

  1. The quality of figure 1a can be improved.

We have provided new Figure 1a with better quality.

  1. Quality of figure 2 can be improved. How many times were this experiment repeated?

We have reworked Figure 2 to address this comment. The experiment was conducted three times and the most common data is shown on the Figure 2.

  1. Figure 3 seems to present unspecific binding. It is not clear the conclusion made by the authors.

We have conducted new experiments on the RAW264.7 cells for all peptides tested and provided new results in the new Figure 3.

  1. The molecular docking can be summarised in one figure, with additional information presented as supplementary material.

We have transferred former Figure 6 to the supplemental and combined former figures 8 and 9 into the one new Figure 7.

  1. The discussion is not aligned with the introduction and results. It can be shortened with clear logic presentation of the findings.

We have made excessive changes to the Discussion Chapter. We have shortened this part of the manuscript by a third removing our suggestions about general mechanisms of proinflammatory receptors activation and other material not directly connected with the main goal of our work. We have added some material to make better connection with obtained results. We have restructured it to remove short paragraphs and make it clearer for the readers.

  1. How was the cytotoxicity % estimated? The authors have not mentioned the negative and positive control used to normalise the data.

The method of cytotoxity testing was added to the beginning of the Results Section to clarify this issue. Cytotoxicity was measured using a Cytotox 96 analysis kit (Promega, Madison, WI, USA) according to the manufacturer protocol. It includes several positive and negative controls for each measured point:

  1. Effector cells-spontaneous LDH Release Control.
  2. A constant number of target cells + a certain number of effector cells.
  3. Target cells for control – Target Cell Spontaneous LDH Release Control.
  4. Target cells for control – Target Cell Maximum LDH Release Control.
  5. Cell medium + lysing solution (10x) – Volume Correction Control.
  6. Cellular environment - Culture medium background control.

  1. The methods sections contain one-sentence paragraphs that can be connected with paragraphs with similar ideas or expanded.

We have improved the Methods Section to address this point.

  1. The title does not reflect its content.

We have changed the Title of the article second time to make it more connected with the content of the manuscript.

Round 3

Reviewer 3 Report

Comments and Suggestions for Authors

I have no further comments. All comments have been addressed.